# Extracellular Vesicles from Plants: Current Knowledge and Open Questions

**DOI:** 10.3390/ijms22105366

**Published:** 2021-05-20

**Authors:** Ornella Urzì, Stefania Raimondo, Riccardo Alessandro

**Affiliations:** 1Department of Biomedicine, Neuroscience and Advanced Diagnostics (Bi.N.D), Section of Biology and Genetics, University of Palermo, 90133 Palermo, Italy; ornella.urzi@unipa.it; 2Institute for Biomedical Research and Innovation (IRIB), National Research Council (CNR), 90146 Palermo, Italy

**Keywords:** plant-derived extracellular vesicles, omics characterization, anti-tumor effects, anti-inflammatory effects, drug-delivery vehicles

## Abstract

The scientific interest in the beneficial properties of natural substances has been recognized for decades, as well as the growing attention in extracellular vesicles (EVs) released by different organisms, in particular from animal cells. However, there is increasing interest in the isolation and biological and functional characterization of these lipoproteic structures in the plant kingdom. Similar to animal vesicles, these plant-derived extracellular vesicles (PDEVs) exhibit a complex content of small RNAs, proteins, lipids, and other metabolites. This sophisticated composition enables PDEVs to be therapeutically attractive. In this review, we report and discuss current knowledge on PDEVs in terms of isolation, characterization of their content, biological properties, and potential use as drug delivery systems. In conclusion, we outline controversial issues on which the scientific community shall focus the attention shortly.

## 1. Extracellular Vesicles from Plants

In the last two decades, there has been an exponential increase in the amount of research aimed at studying the mechanisms of cell–cell communication mediated by extracellular vesicles (EVs). EVs are a family of lipoproteic structures, released by prokaryotic and eukaryotic cells, heterogeneous in terms of origin, size, and content. To date, the interest of the scientific community in the field has been focused on the purification of EVs from animal cells and biological fluids, as well as on their morphological and functional characterization. More recently, the role that these vesicles may have in cross-kingdom communication is attracting the attention of many research groups; in particular, growing studies are focusing on the comprehension of the interactions between vesicles isolated from plants, in this review defined as plant-derived EVs (PDEVs), and mammalian cells. This interest certainly stems from the natural origin of these structures and the potential applications that derive from them, especially in the field of human health. The first study related to the extracellular release of small vesicles in the plant kingdom was from 1967 [1], but it was only later, with the research of Regente [2] on sunflower apoplastic fluid, that the number of studies on PDEVs increased. Starting from the investigations of Zhang et al. in 2013 [3], several studies have focused on EVs isolated from the juice of different fruits; recently, the attention is focused on the characterization of EV content.

In this section, we review the current methods used to isolate PDEVs from different plant matrices as well as the studies aimed at identifying their RNAs, protein, and lipid content. Then, in the next paragraphs, we discuss the data focused on their biological properties and their potential use as drug delivery vehicles.

### 1.1. Isolation Techniques

Although the interest in PDEVs has grown in recent years and many research groups are exploring their properties, a standardized and unique isolation protocol still does not exist.

The most common method of plant vesicle isolation is differential centrifugation followed by ultracentrifugation [4,5,6,7,8,9,10,11,12]. The starting material can be fruits [4,6,7,8,9,10], roots [5,13], stems [9,14], leaves [9,14,15,16], seeds [16], and saps [17]; these matrices can be manually squeezed or using a mixer to obtain the juice. The juice is subjected to several differential centrifugation steps: low-speed centrifugation (about 500–3000× *g* for 10–15 min) to remove plant fibers and large particles; intermediate speed centrifugation (2000–10,000× *g* for 20–40 min) to remove large debris and subcellular organelles; and high-speed centrifugation (100,000–150,000× *g* for 1.5–2 h) to obtain PDEVs pellet.

In addition to these centrifugation steps, some protocols include filtration steps with 0.8-, 0.45-, and 0.22-micron pore size filters. However, the type, quantity, and quality of PDEVs obtained by ultracentrifugation can be influenced by several parameters, such as *g*-force, rotor type, rotor sedimentation angle, and solution viscosity. Furthermore, since ultracentrifugation also sediments other vesicles, proteins, and protein/RNA aggregates, a subsequent sucrose density gradient step is used to separate PDEVs from contaminants [5,9,10,11,12].

Other methods such as ultrafiltration or immune isolation that are routinely used for animal-derived EVs have not been widely used for plant-derived EVs. Sashin et al. used the Exo-spin™ Exosome Purification Kit, which combines precipitation with size exclusion chromatography, and successfully isolated EVs from *Triticum aestivum* [15]. Recently, Yang et al. proposed an alternative method to isolate EVs from the lemon. This method combines electrophoretic technique with a 300 kDa cut-off dialysis bag. They centrifuged lemon juice at 3000× *g* for 10 min and 10,000× *g* for 20 min and filtered the supernatant through a 0.22 μm pore size filter. Then, the juice was placed in a 300 kDa dialysis bag placed in a gel holder cassette with a current of 300 mA [18].

The yield of PDEVs obtained varies depending on the starting material. For example, Raimondo et al., starting from 240 mL of lemon juice, isolated about 600 µg of nanovesicles [4]. Another group, however, obtained about 10 mg of EVs from 10 g of *Dendropanax morbifera* sap [17]. In addition to protein quantification, the recovery of EVs can be determined by other techniques, such as cytofluorimetry. For example, Potestà et al. determined that the number of EVs contained in 1 mg of *Moringa olifera* seed extracts is 16,921 ± 617 [19].

Despite being quite variable, the yield of plant-derived vesicles is higher than those obtained from animal cells; this represents a very attractive point for their potential therapeutic use. However, the lack of a standard isolation method still represents a limitation to their use.

### 1.2. Content Characterization of PDEVs

Considering the complex and heterogeneous content of PDEVs, omics analysis plays a key role in the characterization and identification of their content. Several studies published to date have reported proteomic, lipidomic, metabolomic, and RNA seq analyses of vesicles isolated from various plant species, leading to the identification of proteins or lipids that could potentially serve as markers in the future, and in parallel to define specific molecular profiles of EVs from each species.

#### 1.2.1. Small RNAs in PDEVs

One of the most interesting findings regarding PDEVs content concerns the presence of small RNAs (sRNAs), in particular microRNAs (miRNAs); these complexes may represent a new class of cross-kingdom modulators, by mediating animal–plant interactions at the molecular level [20,21]. Xiao et al. observed the presence of miRNAs in 11 different plant species [22]; subsequently, they analyzed the expression distribution of miRNAs isolated from coconut, orange, and tomato EVs, categorizing them into frequent miRNAs, moderately present miRNAs, and rare miRNAs. Target prediction analyses using TargetScan showed that some of the most expressed miRNAs regulate the expression of mammalian genes associated with the inflammatory and tumor response [22]. miRNAs were also found in EVs from ginger [12]; specifically, some of these miRNAs target several genes from *Lactobacillus rhamnosus*, thereby modulating the composition of the host microbiome [12].

In a study on vesicles isolated from *Moringa oleifera*, 19 miRNAs belonging to 20 conserved families of plant miRNAs were identified, two of which, miR396a and miR396c, were more abundant in vesicles than in seed aqueous extract. The presence of these miRNAs was then correlated with the reduction of viability of tumor cells treated with EVs [19]. Analysis of miRNAs was also performed in strawberry juice and the PDEVs, detecting only miR166g in both matrices [8].

A recent study, through a comparison of sRNA profiles obtained from EVs isolated from Arabidopsis, revealed the presence of small RNAs of 10–17 nucleotides, called “tiny RNAs”, whose function is still unknown [23].

Finally, a preliminary study published in March 2021 demonstrated, through in silico target prediction analysis, that vesicles from soybean, ginger, hamimelon, grapefruit, tomato, and pear possess multiple miRNAs targeting different regions within SARS-CoV-2. If further studies confirm these analyses, PDEVs containing these miRNAs could represent an attractive therapeutic strategy to target altered gene expression related to pathologic conditions [24].

While for the animal kingdom increasing studies are focused on identifying the mechanism of sorting of sRNAs in EVs [25,26], in the plant kingdom this had not been studied until recently; a very recent article has for the first time highlighted how even the small RNAs present in PDEVs are the result of a selective loading process operated by several RNA-binding proteins. In particular, the authors identified in *Arabidopsis thaliana* EVs Argonaute 1 (AGO1) and RNA helicases (RH11 and RH37) that selectively bind sRNAs enriched in EVs but not those not contained in the vesicles [27]. Given the importance of RNAs of PDEVs in modulating gene expression in mammals, a topic that is discussed in the following paragraphs, the in-depth study of the mechanisms of RNA loading in vesicles will assume considerable relevance in the future especially taking into account their possible therapeutic use.

#### 1.2.2. Protein Profile of PDEVs

EVs do not contain a random profile of proteins, but rather the specific protein composition depends on their origin in terms of secretory pathways and matrices. As discussed above, the origin of PDEVs is still debated and depends on the isolation techniques and the plant matrices; however, some protein families have been identified in PDEVs from different species.

One of the families widely found in PDEVs is that of annexins. Annexin A1 and Annexin A2 are crucial in the biogenesis of mammalian EVs and for the formation of the multivesicular bodies [28,29]. In the plant kingdom, these proteins have been identified in EVs isolated from different matrices such as juice and apoplastic fluid. They are found in PDEVs from the juice of four *Citrus* species [4,30] and in those from the apoplastic fluid of sunflower seeds [2]. In addition, they were found in PDEVs from the apoplastic fluids of Arabidopsis leaves (*Arabidopsis thaliana*); these EVs are also enriched in proteins involved in biotic and abiotic stress responses [31].

Mass spectrometry approaches have also allowed the identification of another family of proteins extensively described in EVs from animal cells, the Heat Shock Proteins (HSPs). HSP60 is found in PDEVs from sunflower, as well as HSP70, which has also been described in PDEVs from several citrus fruits [30] where HSP80 and HSP90 have been also found [4,32]. In a recent study of EVs isolated from tomato by size exclusion chromatography techniques, high levels of HSPs were found, together with lipoxygenase and ATPases [33]. Finally, proteins belonging to the Aquaporin family were found in PDEVs from citrus [4,30,34] and grape [3]. Aquaporins were also reported in vesicles isolated from broccoli [35]; in this study, the authors demonstrated that the presence of these proteins is correlated with the stability of the EV plasma membrane and with the osmotic water permeability. Interestingly, in a recent study on EVs isolated from *C. plantagineum* and *N. tabacum*, the authors identified proteins involved in the cell wall remodeling such as hydrolases, e.g., 1,3-β-glucosidases, pectinesterases, polygalacturonases, β-galactosidases, and β-xylosidase/α-L-arabinofuranosidase 2-like [36].

#### 1.2.3. Lipid and Metabolic Profile of PDEVs

The lipidomic analysis of PDEVs has nowadays raised interest because their role in the interaction with mammalian cells, as well as many of the functional effects of these vesicles, can be attributed to this component. The major lipid species found in PDEVs are phosphatidic acid (PA), phosphatidylethanolamine (PE), and phosphatidylcholine (PC). Phospatidic acid was described in the vesicular fraction of sunflower apoplastic fluid [2]; it is enriched in grape-EVs compared to the whole juice [3] and in ginger-derived EVs [12]. In particular, in the last study, the presence of PA was correlated to internalization of ginger EVs by specific intestinal bacteria, Lactobacillus rhamnosus, while the presence of PC to the uptake by intestinal Ruminococcaceae [12].

Phospatidic acid was recently reported in nanovesicles from *Uvae-ursi folium, Craterostigma plantagineum*, and *Zingiberis rhizoma* [36]. Phosphatidylcholine was described in grapefruit [32], together with PE that was also found in grape-EVs [3], and in nanovesicles from *C. plantagineum* [36]. In this last study, PE was detected in EVs from *C. plantagineum* and *Zingiberis rhizoma* [36].

Recently, accurate lipidomic analysis of Arabidopsis rosette leaf EVs and the whole leaf tissues allowed the identification of 23 classes and 279 species of lipids in EVs. Interestingly, the EV-lipid profile showed enrichment in sphingolipids (around 46%) in particular of glycosylinositolphosphoceramides compared to the leaf tissue [37].

In addition to lipid species, increasing studies also include a metabolomic analysis of PDEVs that, in addition to the other EV-enclosed biomolecules, may explain their beneficial properties. In PDEVs from ginger, the phytochemical shogaol was identified [13], while broccoli-derived EVs contain sulforaphane, a compound of the isothiocyanate group [38]. In a study from 2014, the flavonoid naringenin was found in grapefruit-EV [32]. More recently, another group working on grapefruit-derived EVs performed an untargeted GC-MS analysis to identify the metabolites of three different matrices: juice, microvesicles, and nanovesicles. The results from this analysis show that the samples differ in terms of composition; in particular, the juice is enriched in fructose, citric acid, glucose, sucrose, and myo-inositol. Sugars and their derivates were also found in microvesicles, together with quinic and oxalic acid, while the nanovesicles are enriched in organic acids, such as glycolic and citric acids, and amino acids [39]. Ascorbic acid was found in strawberry-derived EVs (416 nmoles/mg EVs) [8].

The schematic representation of the content of plant-derived EVs is shown in Figure 1.

## 2. Biological Properties of PDEVs

Since their preliminary description in the 1960s [1], plant-derived extracellular vesicles (PDEVs) have aroused increasing interest in the field of scientific research. As described in the previous paragraph, different studies highlighted that PDEVs contain functional biomolecules such as proteins, lipids, RNAs, and metabolites, which can mediate cell–cell communication. The physiological role of PDEVs appears to be related mainly to plant immune response [21,31,36] and plant–microbe symbiosis [40,41]. Nevertheless, PDEVs have been shown to interact with mammalian cells, showing remarkable biological properties responsible for a cross-kingdom interaction. In this section, we discuss the studies on anti-tumor, anti-inflammatory, and immune-modulatory activities of PDEVs.

### 2.1. Anti-Tumor Properties

Several studies have emphasized the anti-cancer properties of EVs derived from different plants, thus enabling them to become potential therapeutic compounds in combination with current treatments in cancer management [4,5,14,17,18,19,39].

In 2015, Raimondo et al. isolated EVs from *Citrus limon* juice, with a size of 50–70 nm, which were able to inhibit cell proliferation of three tumor cell lines: A549 (human lung carcinoma), LAMA84 (human chronic myeloid leukemia), and SW480 (human colorectal adenocarcinoma). The arrest in cell proliferation was selective for tumor cells because the same treatment did not affect normal cell growth, and it was mediated by TRAIL (TNF-related apoptosis-inducing ligand) [4]. Further studies from the same group demonstrated that proteins belonging to the lipid metabolism pathway were differentially modulated by lemon EV treatment in colon colorectal adenocarcinoma cell line; among those, Acetyl-CoA carboxylase 1 and phospholipase DDHD1 were downregulated [42]. In line with this evidence, a recent study by the same research group showed that the in vivo administration of a food supplement containing lemon EV, isolated at industrial scale, reduces LDL cholesterol in healthy volunteers [21].

Similarly, another group demonstrated that lemon-derived extracellular vesicles (LDEVs) can be internalized by human gastric cancer cell lines both in 2D cultures of AGD and BGC-823 cells and in a 3D culture of SGC-7901 spheroids. They found that LDEVs- treatment induced GADD45A expression in gastric cancer cells. GADD45A is a protein involved in cell cycle control and DNA repair, and it is considered a tumor suppressor [43]. The treatment of gastric cancer cells with LDEVs suppressed cell growth and induced apoptosis by upregulating GADD45A gene and protein expression and inducing reactive oxygen species (ROS) production [18]. Interestingly, both EVs isolated from *Citrus limon* juice and LDEVs had in vivo anti-tumor activities [4,18]. Moreover, a recently published study analyzed the anti-cancer properties of micro- and nanovesicles (MVs and NVs) derived from four *Citrus* species: *C. sinensis*, *C. limon*, *C. paradisi*, and *C. aurantium*. Both MVs and NVs isolated from all Citrus fruits negatively influenced cell growth of A375 (human melanoma), A549, and MCF-7 (human breast carcinoma) but not of HaCat (human keratinocytes) cells [39]. In particular, MVs and NVs from *C. paradisi* were able to arrest the cell cycle of melanoma cells at the G2/M phase by enhancing the gene expression of p21, a cell cycle inhibitor, and reducing Ciclyn B1 and Ciclyn B2 levels, which regulate G2/M transition [44]; in addition, these EVs promoted apoptosis through activation of PARP-1 [39]. It was found that plant EVs can exert their anti-cancer activity acting on cells that take part in the tumor microenvironment. For instance, EVs derived from *Panax ginseng* (called GDNPs) could induce M1-like polarization in macrophages through the activation of Toll-like receptor (TLR)-4/myeloid differentiation antigen 88 (MyD88) signaling pathway. This polarization was accompanied by an increase in the production of ROS and the conditioned media of GDNPs-treated macrophages were able to induce apoptosis in B16F10 cells (mouse melanoma), by increasing caspase 3/7 protein expression [5]. According to this evidence, EVs derived from *Dendropanax morbifera*, both from leaves and stems, inhibited melanogenesis by reducing the expression of TYR, TRP-1, and TRP-2 in B16BL6 cells (mouse melanoma). These effects were mediated by the suppression of MITF (melanogenesis-associated transcription factor) expression through the UV-dependent α-MSHMC1R pathway in melanoma cells; EVs from leaves were stronger in the TYR inhibition than EVs derived from stems [14]. Another research group has isolated EVs from *Dendropanax morbifera* (DM), *Pinus densiflora* (PD), *Thuja occidentalis* (TO), and *Chamaecyparis obtusa* (CO) saps to test their cytotoxicity on human cancer and normal cells. They found that DM-EVs had cytotoxic effects on breast and skin cancer cells (MDA-MB-231, MCF7, and A431) but not on normal MCF10A (breast cells) and HNF (skin cells) cell lines. PDEVs, instead, decreased cell viability of MCF7 and especially A431 cells; TO-EVs and CO-EVs did not show cytotoxic effects on any cell line. The author demonstrated that the co-treatment with DM-EVs and PDEVs had a synergic effect against tumor cell growth and improved apoptosis, but the mechanisms underlying these results remain unexplained [17]. The same group developed a 3D microfluidic cancer metastasis model to deepen the role of DM-EVs on cancer-associated fibroblasts (CAFs). This model employed a 3D microfluidic device supported by the collagen gel in which human umbilical vein endothelial cells (HUVECs) were seeded as a monolayer; HUVECs were differentiated into CAFs through the treatment with melanoma-derived exosomes, reproducing the tumor microenvironment [45]. DM-EVs caused a decrease in the survival rate of CAFs both when administered with repeated treatment and with pre-and co-treatment. Moreover, DM-EVs-treated CAFs showed a gene expression panel different from that of untreated cells. Among the genes with different expressions, there were cell migration related-genes (TFG-β2, PDGFC, ILK, and AK) and extracellular matrix (ECM)-related genes (CD44, PLAU, COL3A1, COL4A6, ITGA11, and ITGA6) [46]. Recently, Potestà et al. studied the properties of microvesicles (MVs) isolated from *Moringa oleifera* seed aqueous extract (MOES) [19]; in previous work, the same authors demonstrated that MOES contains miRNAs which could be responsible for its anti-proliferative and pro-apoptotic effects on cancer cells, and not on healthy cells [16]. In the recent study [19], researchers treated Jurkat and HeLa cells (respectively, human acute T cell leukemia and cervical adenocarcinoma cells) with MOES MVs for 72 h and observed cytotoxic and pro-apoptotic effects in both tumor cell lines, even if HeLa were more resistant than Jurkat cells. In addition, the same treatment did not affect the cell growth of PBMCs isolated from healthy donors, therefore, the MOES MVs could selectively affect tumor cell proliferation and apoptosis. The pro-apoptotic property of MOES MVs could be attributed to their miRNA content since when the authors treat tumor cells with the *mol*-sR pool, they observed a comparable result to that of MVs [19].

### 2.2. Anti-Inflammatory Properties

Growing evidence has demonstrated that plant-derived EVs can also have anti-inflammatory properties [3,6,9,10,38,47,48]. Inflammation can be the leading cause of many diseases including ulcerative colitis, obesity, diabetes, heart diseases, cancer, and non-alcoholic fatty liver diseases (NAFLD). Considering the side effects of existing anti-inflammatory therapies, the development of new drugs from natural sources is gaining interest among the scientific community [49].

Different groups found that grape exosomes-like nanoparticles (GELNs) have a protective effect against dextran sulfate sodium (DSS)-induced colitis [3,10]. Ju et al. demonstrated that after gavage administration, GELNs were accumulated in the intestinal stem cells (Lgr5-EGFP^+^) of Lgr5-EGFP-IRESCreERT2 mice, enhanced the proliferation of the intestinal epithelium cells, and increased the number of intestinal stem cells by upregulating Sox2, Nanog, OCT4, KLF4, c-Myc, and EGFR gene expression. These results encouraged the authors to investigate the effects of GELNs in mice with DSS-induced colitis. GELNs decreased mortality and contrasted the reduction of the intestine length in DSS treated mice by inducing the gene expression of Lgr5 and BMI1, two markers of intestinal stem cells [50], and the nuclear translocation and activation of β-catenin in intestinal crypt cells [3].

In line with these findings, another group analyzed the biological function of different edible plant-derived exosomes-like nanoparticles (EPDENs) and found that when murine macrophages (RAW264.7) were treated with EPDENs isolated from ginger, the gene and protein expression of heme oxygenase 1 (HO-1) and interleukin 10 (IL-10) was upregulated; carrot EPDENs, instead, upregulated only IL-10 levels [47]. HO-1 and IL-10 expression in macrophages is essential to prevent colitis since they have an anti-inflammatory function [51,52,53,54]. The upregulation of HO-1 and IL-10 was explained by the nuclear translocation of NRF2 in RAW264.7 treated with ginger EPDENs; it is known that NRF2 can stimulate HO-1 and IL-10 activation [55]. This, accompanied by the ability of these nanoparticles to reach intestinal macrophages when orally administrated to mice, makes them possible candidates for colitis treatment [47]. Moreover, among ginger-derived nanoparticles (GDNPs), a subpopulation called GDNPs 2 has been identified; GDNPs 2 possesses beneficial properties towards acute colitis and could prevent chronic colitis and colitis-associated cancer (CAC). In particular, it was found that GDNPs 2 treatment could reduce lipocalin-2 (Lcn-2) fecal levels in mice with DSS-induced colitis, suggesting an anti-inflammatory role of these nanoparticles; Lcn-2 is considered a biomarker for intestinal inflammation [56]. GDNPs 2 reduced spleen weight and contrasted the reduction of colon length decreasing pro-inflammatory cytokines, such as tumor necrosis factor-α (TNF-α), interleukin 6 (IL-6), and interleukin 1-β (IL-1β), and increasing anti-inflammatory cytokines, e.g., interleukin 10 (IL-10) and interleukin 22 (IL-22). Similar results were obtained in IL10 knockout (IL10^−/−^) mice, using a chronic colitis model, in which GDNPs 2 were able to prevent splenic enlargement and colon length reduction, by downregulating TNF-α and IL-1β gene expression. Finally, the administration of GDNPs 2 to AOM/DSS mice treated with the carcinogen Azoxymethane and DSS (AOM/DSS group), which recapitulate CAC, decreased Lcn-2 levels, the number of tumors per mouse, and IL-6 and IL-1β expression compared with the AOM/DSS group [48].

Another source of EVs which has been demonstrated to have protective properties against colitis in mice is broccoli [38]. Broccoli-derived nanovesicles (BDNs) were able to contrast the increase of pro-inflammatory cytokines, such as TNF-α, IL-17A, and IFN-γ, in colonic tissues of two colitis models (DSS-induced and T cell transfer model of colitis). Among the targets of BDNs, there were dendritic cells (DCs), whose number was reduced in BDN-treated mice compared with the control group. DCs treated with BDNs presented a tolerogenic signature, since they had higher expression levels of TGF-β, interleukin 10 (IL-10), and Aldh1a2; moreover, BDNs inhibited the recruitment of monocytes into the inflamed colon by decreasing chemotactic chemokines (CCL2, CXCL1, and CCL20). In vitro experiments showed that the treatment of BMDCs with lipids, especially sulforaphane (SFN), derived from BDNs induced a tolerogenic phenotype by activating AMPK [38].

Cheng et al. investigated the effects of several plants (cilantro, aloe vera, grapefruit, garlic, turmeric, dandelion, lavender, cactus, and ginger) derived EVs on NLRP3 inflammasome activation, a biological process involved in the initiation and progression of autoinflammatory, neurodegenerative, and metabolic diseases. Only ELNs isolated from ginger showed, in different murine macrophage cells, the ability to prevent NLRP3 inflammasome activation by inhibiting IL-1β and interleukin 18 (IL-18) release and Casp1 p10 protein expression levels, markers of this biological process [57]. Besides, ginger-derived ELNs suppressed the oligomerization of the apoptotic speck protein containing a caspase recruitment domain; PDEV activity seems to be attributed to their lipid content [9]. Recent evidence shows that blueberries-derived ELNs (B-ELNs) also have anti-inflammatory properties on human endothelial cells (EA.hy926) [6]. Pre-treatment of EA.hy926 cells with B-ELNs was able to revert TNFα-induced cell death as well as ROS production. It was found that B-ELNs downregulated the gene expression of IL-6, interleukin 1 receptor-like 1 (IL1RL1), mitogen-activated protein kinase 1 (MAPK1), intercellular adhesion molecule 1 (ICAM1), toll-like receptor 8 (TLR8), and TNF and upregulated the gene expression of heme oxygenase 1 (HMOX1) and nuclear respiratory factor 1 (NRF1). These anti-inflammatory and antioxidant effects could be attributed to their miRNA content, since B-ELNs contain miR-156e, miR-162, and miR-319d, which potentially target PTGIS, MAPK14, and PDE7A genes [6]. It was recently found that strawberry-derived EPDENs have antioxidant properties as well. Adipose-derived mesenchymal stem cells (ADMSCs) pre-treated with strawberry EPDENs and then stimulated with H_2_O_2_ showed reduced cell death and lower ROS levels than cells treated with H_2_O_2_ alone, probably because EPDENs contain vitamin C [8].

Ginger ELNs also possess biological effects on hepatocytes, as it was seen that they can prevent alcohol-induced injury. These vesicles were internalized by primary murine hepatocytes and led to nuclear translocation of Nrf2 through TLR4/TRIF pathway. Furthermore, upregulation of antioxidant and detoxifying genes, such as HO-1, NQO1, GCLM, and GCLC, was observed in the liver of ginger ELNs-treated mice accompanied by protection from alcohol-induced damage, as the vesicles reduced ROS, liver triglycerides, and liver weight compared to alcohol-only mice [13]. Recently, was found that orange juice-derived nanovesicles (ONVs) could ameliorate obesity [7]. The treatment with ONVs of a co-culture of CACO and HT29 cells used as a model of in vivo intestinal barrier (IBs), decreased triglycerides and promoted their release in association with chylomicrons. These in vitro data were confirmed by in vivo experiments in which HFHSD (high-fat, high-sucrose diet) mice were treated with ONVs. The vesicles accumulated primarily in the jejunum of mice and induced an increase in villus size. ONVs gavage decreased triglyceride levels in the jejunum, chylomicron release, and gene expression of ANGPTL4, a novel therapeutic target of colonic inflammation.

Lastly, a work published in 2018 showed that plant EVs may also play a role in skin regeneration in vitro. It was found that nanovesicles isolated from *Triticum aestivum* (wheat) increased cell proliferation and migration of three cell lines: human dermal fibroblasts (HDFs), HUVECs, and HaCaT. In addition, wheat nanovesicles showed pro-angiogenic effects, as they promoted tube formation in HUVECs and increased COL1A gene and protein expression in HDFs [15].

### 2.3. PDEVs Modulate Mammalian Microbiota

The microbiota is the set of symbiotic microorganisms that coexist with the human organism without damaging it; the diet can modulate its composition while its alterations can lead to the onset of diseases [58]. Some recent evidence shows how PDEVs can interact with the microbiota and the result of this cross-talk may offer new opportunities for their use as potential therapeutic agents [11,12,59]. Ginger-derived ELNs administration in mice led to a modification of gut microbiota, leading to the increase of Lactobacillaceae and Bacteroidales S24-7 and the decrease in Clostridiaceae with respect to the control group. Ginger ELNs were preferentially internalized by *Lactobacillus rhamnosus* (LGG) due to their phosphatidic acid (PA) lipids and promoted their growth by repressing LexA gene and protein expression. The small RNAs contained in ginger ELNs had protective effects against DSS-induced colitis by reducing the levels of pro-inflammatory cytokines, such as IL-1β and TNF-α, and by promoting the expression of interleukin 22 (IL-22) through I3A, which activates the aryl hydrocarbon receptor pathway [12]. The physiological role of ELNs is to defend plants from infection; however, in 2019, Sundaram et al. showed that ELNs can also protect mammalian cells against pathogens. Specifically, they demonstrated that ginger-derived ELNs were selectively internalized by *Porphyromonas gingivalis* and inhibited its growth. Ginger ELNs increased membrane depolarization, a key factor in the regulation of viability and signal transduction pathway in bacteria, through their lipids, particularly PA (34:2). The 34:2 PA was able to interact with a membrane protein of *Porphyromonas gingivalis*, hemin-binding protein 35 (HBP35), which is essential for the growth and survival of this bacterium [60]. Lipids and miRNAs from ginger ELNs also inhibited the expression of *Porphyromonas gingivalis* virulence-related genes, such as AraC, HagA, and OmpA, as well as bacterial attachment and invasion of gingival epithelial cells (TIGKs). These results were confirmed by in vivo experiments showing that administration of the ginger ELNs reduced *P. gingivalis* colonization in the oral cavity of mice and bone loss [11]. The same group investigated the effects of lemon ELNs in *Clostridioides difficile* (C. difficile) infection. The authors treated C diff-infected mice with a probiotic mixture containing *Lactobacillus rhamnosus GG* (LGG) and *Streptococcus thermophilus ST-21* (STH) (LS) pre-treated with LELNs (LELN-LS) and observed that colonic length was similar to that of uninfected mice and that mortality was reduced compared with LS treatment alone. Metabolomic analysis and HPLC showed that I3Ad and I3LA, two ligands of the AhR, were increased in the group that received LELN-LSs compared with the PBS and LS groups. Treatment with LELN-LS decreased the colony-forming unit of C difficile in the feces of mice compared with the PBS control group since the treatment induced an increase in intestinal lactic acid, which leads to a decrease in indole production by inhibiting the gene expression of tryptophanase tnaA, responsible for indole synthesis [59].

The main findings discussed in this section are summarized in Figure 2.

## 3. PDEVs as Drug Delivery Vehicles

The research and development of new drugs is a primary need in the world since, for many diseases, an effective cure has not yet been found. The main problems with conventional drug therapies are: (i) poor selectivity; and (ii) difficulty in crossing biological barriers. Chemotherapy, for example, is still the main therapy against cancer; however, its poor selectivity causes several side effects. On the other hand, the blood–brain barrier (BBB), which separates the blood from the extracellular fluid of the central nervous system (CNS), hinders drug delivery from the blood to the brain tissue. In recent years, new drug delivery systems are being explored to overcome these challenges. Extracellular vesicles possess several characteristics that make them suitable as drug delivery systems, such as the ability to cross biological barriers, the stability in the circulatory system, and safety; indeed, several studies investigated the possibility to use them as new drug nanocarriers [61,62]. However, some limitations to the use of EVs as drug delivery vehicles still exist as the development of a scalable and reproducible EV isolation method and the immune response that its administration may trigger.

PDEVs represent a promising model for drug delivery, as they are natural products that possess several advantageous properties including safety, non-toxicity, low immunogenicity. Moreover, PDEVs can be produced on large scale and several studies have demonstrated that they are stable and resistant in the stomach- and intestinal-like solutions [10,13,32,48]; they have been found, after oral gavage, in intestinal stem cells, liver, colon, and dendritic cells as well as in the intestinal macrophages of mice [3,10,13,32,38,48]. Therefore, in recent years, some studies have focused on the employment of PDEVs or their derived lipids as drug delivery vehicles.

Particularly, in 2013, Wang et al. developed nanovectors using lipids derived from grapefruit EVs and named them grapefruit-derived nanovectors (GNVs) [63]. GNVs were successfully internalized by different cell types, including GL26 (mouse glioma cells), A549 (human lung carcinoma cells), SW620 (human colon carcinoma cells), CT26 (mouse colon carcinoma cells), and 4T1 (mouse breast tumor cells), without inducing cytotoxicity. The authors evaluated the bio-distribution of GNVs following three different routes of administration and observed that after tail-vein or intraperitoneal injections the GNVs were located in the liver, lung, kidney, and splenic tissue, while GNVs were found in the lung and brain after intramuscular and intranasal administration. They have also shown that GNVs can carry specific therapeutic agents, such as DNA, proteins, and short interfering RNAs (siRNAs), and can be modified to specifically target tumor tissues. As a proof of concept, they demonstrated that GNVs carrying Pentoxifylline (PTX), an anti-cancer drug [64], and binding folic acid (FA), which was chosen because many tumors have a high expression of folate receptors [65], were able to target and decrease tumor growth in in vivo model [63].

The same group further demonstrated that ginger EVs conjugated to methotrexate (MTX), an immunosuppressive and anti-inflammatory drug [66], can contrast DSS-induced colitis in mice. After oral administration, ginger EVs conjugated with MTX (GMTX) preferentially localized in macrophages of lamina propria and reduced body weight, colon length shortening, and colon tissue damages in treated mice with respect to the control groups (MTX alone and PBS groups). GMTX decreased gene and protein expression of pro-inflammatory cytokines, such as TNF-α, IL-1β, and IL-6, in intestinal macrophages and showed fewer undesirable effects than MTX alone [32]. They also developed a novel siRNA transport system using nanocarriers made of lipids derived from ginger EVs, called GDLVs. Once they isolated lipids from ginger ELNs, they reassembled them into nanoparticles using a standard method based on the hydration of a lipid film. GDLVs did not induce cytotoxicity in either RAWs or CT26 cells and were safe in vivo following oral administration. The authors encapsulated siRNA-CD98 in GDLVs and demonstrated that this siRNA carried by nanoparticles inhibited CD98 expression both in vitro, in RAWs and CT26 cells, and in vivo, in the ileum and colon of mice compared to the control group [67]. Furthermore, in 2015 they generated GNVs coated with inflammatory chemokine receptor enriched membrane fraction of activated T cells, called IGNVs. IGNVs had a greater ability to migrate through a monolayer of HUVECs than GNVs and to home in sites of inflammation in several in vivo inflammatory models (LPS induced skin inflammation, DSS induced colitis model, CT26 colon cancer, and 4T1 breast cancer models). Next, the authors loaded IGNVs with doxorubicin (IGNVs-DOX), a drug widely used in cancer treatment [68], and observed that intravenous injection of IGNVs-DOX into tumor-bearing mice caused higher DOX accumulation in the tumors and lower in the liver than DOX-NP^TM^ and GNVs-DOX. In addition, treatment with IGNVs-DOX inhibited tumor growth and increased survival of the mice compared with the control groups. Similarly, the injection of IGNVs loaded with curcumin (IGNVs-Cur), a known anti-inflammatory agent [68], increased the survival rate of mice with DSS-induced colitis and decreased TNF-α, IL-6, and IL-1β levels in colonic tissue compared with free Cur and GNVs-Cur [69]. The intranasal administration of GNVs, on the other hand, was shown to be useful for brain delivery [70]. The authors demonstrated that GNVs coated with FA, carrying miR17 (FA-GNV/miR17), successfully transported miR17 to brain tumor of mice and had therapeutic effects since FA-GNV/miR17 prolonged survival of tumor-bearing mice compared with control groups (FA-GNV/miRNA scramble and PBS). FA-GNV/miR17 also increased the number of DX5^+^NK cells in the brain tumor, probably because miR17 inhibited MHCI expression in cancer cells, which promoted NK cell activation [70].

Another group has investigated the drug delivery potential of GNVs, using these vesicles as nanovectors to transport miR-18a, a tumor suppressor microRNA [71], to contrast liver metastasis. GNVs loaded with miR-18a (GNV-miR-18a) were injected in the tail vein of metastatic colon tumor-bearing mice and were internalized by Kupffer cells (KCs); GNV-miR-18a inhibited liver metastasis by inducing M1 macrophages and inhibiting M2 in mice liver. These effects were mediated by the IFNγ/IRF2 axis, which activated interleukin 12 (IL-12); this cytokine was responsible for the induction of natural killer (NK) and natural killer T (NKT) cells that inhibited colon cancer liver metastasis [72]. Finally, ginger-derived exosomes such as nanovesicles (GDENs) coated with FA demonstrated to be able to target and deliver survivin siRNA to tumor sites in vivo. Survivin siRNA was chosen because its gene silencing is effective in the inhibition of tumor growth and metastasis [73]. FA-GDENs carrying survivin siRNA displayed good biocompatibility; they did not alter cell viability of somatic cell (HEK293), Raw 264.7, and cancer cell (KB); and following retro-orbital IV injection they inhibited tumor growth by reducing survivin expression in tumor tissue compared to the scramble group and GDENs without FA [74].

The main findings discussed in this section are summarized in Figure 3.

## 4. Conclusions, Open Questions, and Challenges

In the last two decades, significant growth of studies on extracellular vesicles was observed; it is only in the last years that this increase has regarded PDEVs. The rising evidence on their properties, related to their complex content, together with the possibility to use PDEVs as delivery systems of other therapeutic substances, makes the study of these structures very attractive. Moreover, their natural origin, as well as the possibility of isolating PDEVs from large volumes, represent major advantages for their use in nutraceuticals. Although the industrial application of PDEVs seems to be easier and more rapid than the use of those from the animal kingdom, many efforts still have to be made. Similar to the need to create guidelines for those working with mammalian EVs [75], the same requirement arises for PDEVs; several points remain to be explained. Among these, one of the questions to be clarified is their origin, especially concerning those isolated from the juice. Indeed, while some studies of vesicles isolated from apoplastic fluid, such as those from sunflower seeds [2,76], have investigated their extracellular nature, no information to date is available about the origin of those isolated from plant matrices after squeezing.

The isolation methods, which are not uniform to date, may contribute to the heterogeneity of PDEVs, even when they are isolated from the same plant matrix. Indeed, although the results of the studies discussed in the previous paragraphs are consistent with conferring similar biological properties to PDVEs, often the application of different isolation protocols, as well as the dimensional characteristics of the isolated vesicles, suggests various populations whose content could also differs.

Finally, although omics analyses related to their content are increasing, the absence of specific markers still hampers the characterization of PDEVs.

Considering the industrial application of PDEVs, clinical studies should be carried out to validate their safety, stability, and efficacy in vivo. This will subsequently require an in-depth analysis of the regulatory framework; in fact, how does the use of PDEVs as nutraceuticals fit into the regulatory context? Since PDEVs are structures containing different plant compounds already described as botanicals, can we classify them as food supplements or should we refer to them as novel food?

Answering all of these points certainly requires the effort of researchers in the field, but the resulting findings would support the rapid application of PDEVs in daily use.

## Figures and Tables

**Figure 1 ijms-22-05366-f001:**
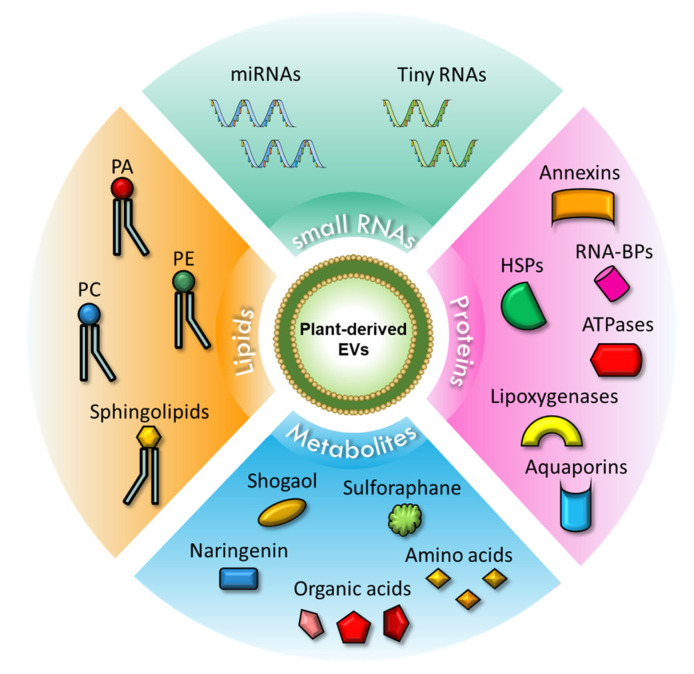
Schematic representation of PDEVs content. Upper corner: Small RNAs contained in PDEVs, which include miRNAs and tiny RNAs. Right corner: Proteins carried in PDEVs, including annexins, RNA-BPs, HSPs, ATPases, lipoxygenases, and aquaporins. Lower corner: Metabolites present in PDEVs, such as shogaol, sulforaphane, naringenin, organic acids, and amino acids. Left corner: Lipids found in PDEVs include PA, PE, PC, and sphingolipids. Abbreviations: miRNAs, microRNAs; HSPs, heat shock proteins; RNA-BPs, RNA binding proteins; PA, phosphatidic acid; PE, phosphatidylethanolamine; PC, phosphatidylcholine.

**Figure 2 ijms-22-05366-f002:**
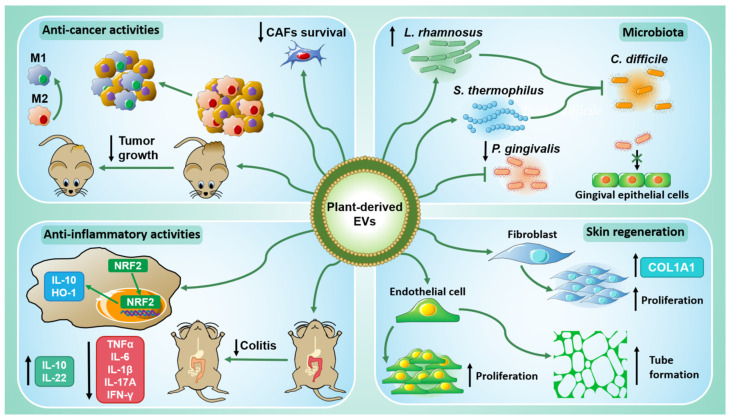
The biological properties of plant-derived EVs. PDEVs have shown anti-cancer activities both in vitro and in vivo (**top left**): they can act directly on tumor cells but also on those of the tumor microenvironment, thus promoting M2 macrophage polarization into M1 macrophages and inhibiting cancer-associated fibroblasts (CAFs). PDEVs have inflammatory activities (**bottom left**) since they upregulate anti-inflammatory cytokines, such as IL-10 and IL-22, and downregulate pro-inflammatory cytokines, TNFα, IL-6, IL-1β, IL-17A, and IFN-γ. They can alleviate colitis in vivo and induce NRF2 nuclear translocation in murine macrophages, leading to IL-10 and HO-1 expression. PDEVs participate in skin regeneration (**bottom right**) by promoting the proliferation and the tube formation of endothelial cells. They can enhance fibroblasts proliferation and upregulate COL1A1 expression. Finally, PDEVs interact also with mammalian microbiota (**top right**) inducing the growth of *Lactobacillus rhamnosus* (*L. rhamnosus*) and inhibiting that one of *Porphyromonas gingivalis* (*P. gingivalis*). Moreover, *L. rhamnosus* and *Streptococcus thermophilus* (*S. thermophilus*) pre-treated with PDEVs can counteract *Clostridioides difficile* (*C. diff*) infection.

**Figure 3 ijms-22-05366-f003:**
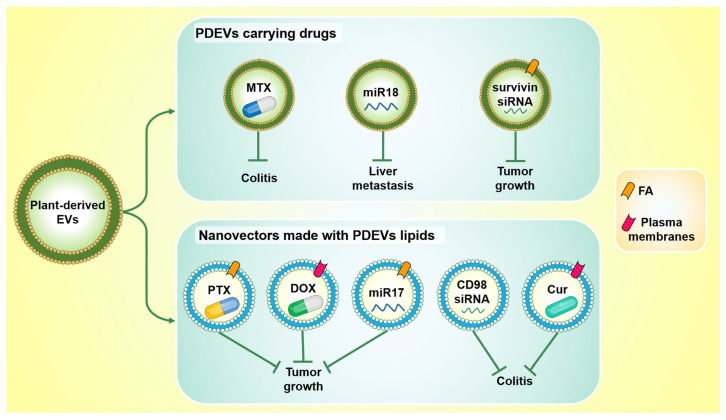
Plant-derived EVs and nanovectors made with their lipids represent promising drug delivery systems. PDEVs can be loaded with both drugs and oligonucleotides (**top**): PDEVs carrying methotrexate (MTX) counteract ulcerative colitis in vivo; PDEVs carrying miR18 reduce liver metastasis in vivo; PDEVs conjugated with folic acid (FA), and loaded with survivin siRNA, inhibit tumor growth. Nanovectors derived from PDEVs lipids have also been shown to be useful as drug delivery vehicles (**bottom**): they inhibit tumor growth when they carry Pentoxifylline (PTX) or miR17 and are conjugated to FA, as well as when they carry doxorubicin (DOX) and are conjugated with inflammatory chemokine receptor enriched membrane fraction (plasma membranes). Moreover, they are also able to counteract ulcerative colitis when they deliver CD98 siRNA or curcumin (Cur) and are conjugated with plasma membranes.

## Data Availability

Not applicable.

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
