# Peer review of "Extracellular Vesicles from Plants: Current Knowledge and Open Questions"

_ijms, 2021, doi:10.3390/ijms22105366_

Round 1

Reviewer 1 Report

Nice work, only some notes:

Line 98 miRNAs abbreviation without defining

Line 487 siRNAs abbreviation without defining

Line 162 , 463 Dont start sentence with abbreviation, in this case use the full name (need to check across the full text)

Line 239 ’D. morbifera’ first, define the ’D’ abbreviation + Line 245 as well. No huge amount of plant names, so please use the scientific name.

Author Response

Palermo, May 17th, 2021

To the Editorial Board of the International Journal of Molecular Science,

We are now submitting a revised version of the manuscript entitled entitled “Extracellular vesicles from plants: current knowledge and open questions” by Urzì, Raimondo and Alessandro for publication in IJMS, together with point-by-point answers to the reviewers. We are very grateful for the opportunity to submit the revised version thus improving the quality of the work and we would like to thank the Reviewers for their relevant and valuable criticisms.

We ask you to reconsider our manuscript for publication in IJMS.

We confirm that this manuscript has not been published elsewhere and is not under consideration by another journal. All authors have approved the manuscript and agree with its submission to IJMS and have declared that no competing interest exists.

Reviewer 1

Nice work, only some notes:

Line 98 miRNAs abbreviation without defining

Line 487 siRNAs abbreviation without defining

Line 162 , 463 Dont start sentence with abbreviation, in this case use the full name (need to check across the full text)

Line 239 ’D. morbifera’ first, define the ’D’ abbreviation + Line 245 as well. No huge amount of plant names, so please use the scientific name.

Reply: We thank the Reviewer for all his/her valuable comments and for appreciating our study. All suggested changes have been made and are highlighted in the manuscript.

Reviewer 2

The present review paper deals with the isolation, characterization, biological properties and potential use as drug delivery systems of plant-derived vesicles. Such a topic addresses the ever increasing interest in the unravelling of the beneficial properties of natural substances e.g. extracellular vesicles released by different organisms, in particular from animal cells.

The review is in most of its parts well written and properly detailed. Furthermore, the authors attempted to address some questions that are still open on the topic, “pushing” the scientific community to deepen their studies in the nearest future.

Reply: We thank the Reviewer for all his/her valuable comments and for appreciating our study. All suggested changes have been made and are highlighted in the manuscript.

My main concerns are:

English can be somewhat improved

Reply: some part of the manuscript has been linguistically revised

Section 1.2 concerning the content characterization of plant-derived extracellular vesicles is too maigre. The authors need to deepen this part including at least a Figure and reporting references on proteomic, lipidomic, metabolomic and RNA analyses of vesicles isolated from various plant species.

Reply: section 1.2 has been improved by adding some references and a schematic figure of plant vesicle contents.

The authors need to report in Section 1 the time-frame related to the previous studies conducted on isolation, characterization, biological properties of such molecules

Reply: timing of previous studies has been added.

Reviewer 3

Specific comments:

1) Line 67: please add full stop after “al” and in all cases through the whole text;

2) Line 70: please add full stop after al. and please remove “derived”;

3) Line 94: please correct “specie” to “species”;

4) Line 100: please remove full stop between [17] and [18]; “Xiao and colleagues” – the citation style should be the same thought the whole text that is Xiao et al.;

5) Lines 109-110: “…authors identified,…….” it should be: “19 miRNAs belonging to 20 conserved families of plant miRNAs were identified”;

6) Line 111: “than in seeds aqueous extract…” – than in seed aqueous extract….”;

7) Line 161: “Phosphatidic” please change to “phosphatidic”;

8) How in the review article unpublished data could be reviewed?

9) Latin names of plant species should be written using Italic font.

10) According to journal requirements references should be cited in order of appearance;

11) “Phosphatidic acid” should be “phosphatidic”;

12) On Figure 2 there is a spelling mistake: “ growht”;

Reply: We thank the Reviewer for all his/her valuable comments and for appreciating our study. All suggested changes have been made and are highlighted in the manuscript. The unpublished manuscipt (indicated in the comment number 8) was recently published, the reference was included in the revised version of this manuscript.

Reviewer 2 Report

The present review paper deals with the isolation, characterization, biological properties and potential use as drug delivery systems of plant-derived vesicles. Such a topic addresses the ever increasing interest in the unravelling of the beneficial properties of natural substances e.g. extracellular  vesicles  released  by different  organisms,  in  particular  from  animal  cells.

The review is in most of its parts well written and properly detailed. Furthermore, the authors attempted to address some questions that are still open on the topic, “pushing” the scientific community to deepen their studies in the nearest future.

My main concerns are:

  • English can be somewhat improved
  • Section 1.2 concerning the content characterization of plant-derived extracellular vesicles is too maigre. The authors need to deepen this part including at least a Figure and reporting references on proteomic, lipidomic, metabolomic and RNA analyses of vesicles  isolated from various plant species.
  • The authors need to report in Section 1 the time-frame related to the previous studies conducted on isolation, characterization, biological properties of such molecules

Author Response

(The authors gave the same response as above.)

Reviewer 3 Report

Manuscript entitled: “Extracellular vesicles from plants: current knowledge and open questions” review methods leading to isolation of plant-derived vesicles, their composition and application, especially selective properties against cancer cells.

Specific comments:

1) Line 67: please add full stop after “al” and in all cases through the whole text;

2) Line 70: please add full stop after al. and please remove “derived”;

3) Line 94: please correct “specie” to “species”;

4) Line 100: please remove full stop between [17] and [18]; “Xiao and colleagues” – the citation style should be the same thought the whole text that is Xiao et al.;

5) Lines 109-110: “…authors identified,…….” it should be: “19 miRNAs belonging to 20 conserved families of plant miRNAs were identified”;

6) Line 111: “than in seeds aqueous extract…” – than in seed aqueous extract….”;

7) Line 161: “Phosphatidic” please change to “phosphatidic”;

8) How in the review article unpublished data could be reviewed?

9) Latin names of plant species should be written using Italic font.

10) According to journal requirements references should be cited in order of appearance;

11) “Phosphatidic acid” should be “phosphatidic”;

12) On Figure 2 there is a spelling mistake: “ growht”;

Author Response

(The authors gave the same response as above.)
